# Understanding drivers of family planning in rural northern India: An integrated mixed-methods approach

Mokshada Jain[1], Yael Caplan[1], B. M. Ramesh[2], Shajy Isac[2,3], Preeti Anand[2,4], Elisabeth Engl[1], Shiva Halli[2], Hannah Kemp[1], James Blanchard[2], Vikas Gothalwal[2,4], Vasanthakumar Namasivayam[2], Pankaj Kumar[5], Sema K. Sgaier[1,6,7]*

**1** Surgo Foundation, Washington, District of Columbia, United States of America, **2** Centre for Global Public Health, Department of Community Health Sciences, University of Manitoba, Winnipeg, Manitoba, Canada, **3** India Health Action Trust, New Delhi, Delhi, India, **4** India Health Action Trust, Lucknow, Uttar Pradesh, India, **5** National Health Mission, Government of Uttar Pradesh, Lucknow, Uttar Pradesh, India, **6** Department of Global Health & Population, Harvard T.H. Chan School of Public Health, Boston, Massachusetts, United States of America, **7** Department of Global Health, University of Washington, Seattle, Washington, United States of America

* semasgaier@surgofoundation.org

**Data Availability Statement:** De-identified datasets used in this paper can be made available upon request. The IRB board, Sigma IRB, New Delhi that approved all the studies directed the de-identified

## Abstract

### Background

Family planning is a key means to achieving many of the Sustainable Development Goals. Around the world, governments and partners have prioritized investments to increase access to and uptake of family planning methods. In Uttar Pradesh, India, the government and its partners have made significant efforts to increase awareness, supply, and access to modern contraceptives. Despite progress, uptake remains stubbornly low. This calls for systematic research into understanding the 'why'—why people are or aren't using modern methods, what drives their decisions, and who influences them.

### Methods

We use a mixed-methods approach, analyzing three existing quantitative data sets to identify trends and geographic variation, gaps and contextual factors associated with family planning uptake and collecting new qualitative data through in-depth immersion interviews, journey mapping, and decision games to understand systemic and individual-level barriers to family planning use, household decision making patterns and community level barriers.

### Results

We find that reasons for adoption of family planning are complex–while access and awareness are critical, they are not sufficient for increasing uptake of modern methods. Although awareness is necessary for uptake, we found a steep drop-off (59%) between high awareness of modern contraceptive methods and its intention to use, and an additional but smaller drop-off from intention to actual use (9%). While perceived access, age, education and other demographic variables partially predict modern contraceptive intention to use, the

datasets should not publicly be made available as they contain sensitive and largely qualitative information on people's private matters around sexual practices and family planning. However, the board has allowed interested individuals to send data requests to the point of contact on the Sigma IRB ethics committee member secretary Dr. U V Somayajulu (Tel: +91 11 4619 555 Email: somayajulu.uv@sigma-india.in Address: Sigma Research and Consulting C 23, South Extension I, Second Floor New Delhi 110 049. India).

**Funding:** This study was funded by Surgo Foundation, a non-profit organization. The funder provided support in the form of salaries for authors [MJ, YC, EE, HK, SS], and senior leadership at the funding organization were involved in study design, data collection and analysis, decision to publish, and preparation of the manuscript. The funder provided grants to the University of Manitoba [BR, SH, JB, VG, VN] which operates the India Health Action Trust [SI, PA, VG] for support with design of research instruments, implementation, and data analysis. The specific roles of all authors are articulated in the 'author contributions' section.

**Competing interests:** The authors have declared that no competing interests exist.

qualitative data shows that other behavioral drivers including household decision making dynamics, shame to obtain modern contraceptives, and high-risk perception around side-effects also contribute to low intention to use modern contraceptives. The data also reveals that strong norms and financial considerations by couples are the driving force behind the decision to use and when to use family planning methods.

## Conclusion

The finding stresses the need to shift focus towards building intention, in addition to ensuring access of trained staff, and commodities drugs and equipment, and building capacities of health care providers.

## Introduction

Around the world, governments and partners have prioritized investments to increase access to and uptake of family planning methods. There is a wealth of evidence linking family planning (FP) to reductions in maternal mortality by reducing the likelihood of unplanned pregnancies, unsafe abortions, and the potential health risks of high parity and closely spaced pregnancies [1]. These benefits are also well documented for children: shorter birth intervals are associated with increases in child mortality risk [2]. By allowing women to better space or limit childbearing and therefore exercise control over family size, family planning also results in better educational and employment outcomes for women as well as improved health and nutrition for children [3]. The importance of family planning is captured in many of the United Nations Sustainable Development Goals' targets for 2030: 3.1 reducing the global maternal mortality ratio, 3.7 ensuring universal access to sexual and reproductive healthcare services, and 5.6 universal access to sexual and reproductive health and reproductive rights [4]. Its importance is similarly reflected in the Family Planning 2020 initiative (FP2020), launched in 2012, which united governments, donors, the private sector, and other organizations to set ambitious FP targets with an ultimate goal of enabling 120 million additional women and girls to use voluntary modern contraception by 2020 [5].

Existing research exploring the drivers and barriers of family planning uptake may explain some of the reasons why women do not use family planning methods, particularly modern contraceptives. Analysis of Demographic and Health Surveys in 52 countries between 2005 and 2014 revealed that the most common reasons for not using contraception despite wanting to limit (not wanting another child) or space (not wanting a child soon) child bearing were fear of side effects, infrequent sex, and opposition to contraception from self or others [6]. In contrast, lack of awareness, lack of access, and cost were rarely reasons for unmet contraceptive need [6]. Several additional studies across low- and middle-income countries also found that fear of side effects, particularly infertility, is a significant barrier to modern contraceptive use [6–9]. Therefore, family planning counselling services and investments in behavior change communication campaigns will be critical to address these concerns and generate demand for modern methods. Women have limited autonomy over their reproductive decisions. Research has identified men as key decision makers and targets for messaging [8,10]. Husbands and relatives, in particular mothers-in-law, heavily influence the fertility decisions made by women in regard to number of sons and timing of sterilization, though norms have begun to shift with young couples making their own contraceptive choices [11]. Some of these findings hold across multiple geographies, suggesting that drivers and barriers of family planning use are

varied and complex, indicating a need for more systematic research to develop a deeper, holistic understanding of what drives family planning decisions. However, to date, no existing study looks holistically at the potential barriers and drivers of use, instead research has focused on a single stakeholder or a limited set of barriers. Holistic studies can help elucidate novel programmatic recommendations as demonstrated by our previous work in Uttar Pradesh around reproductive, maternal, newborn, and child health in government public facilities [12,13]. Our studies used comprehensive frameworks to understand the contextual and internal behavioral drivers shaping provider behavior at facilities and households' care seeking behavior for institutional delivery. We saw an opportunity to apply a similar, holistic framework toward family planning in Uttar Pradesh.

With a population of 1.35 billion and a total fertility rate (TFR) of 2.2 (with some state level TFRs as high as 3.3), India is a key contributor to the FP2020 goals [14]. While female sterilization remains the most popular contraceptive method, in 2017 the Indian Government introduced new contraceptive options–injectable contraceptives and centchorman–to encourage spacing using modern methods [15]. In the most recent National Family Health Survey conducted in 2016, 18.1% of currently married women age 15–49 in Uttar Pradesh reported an unmet need for family planning [16]. As India's largest state, Uttar Pradesh has undertaken many continued efforts to date to increase its modern contraceptive prevalence rate (mCPR). These efforts include increasing the number of IUCD and sterilization delivery points, building the capacity of healthcare providers through training in injectables and IUCDs, improved identification and support of couples with unmet need, and ensuring the availability of trained staff, commodities, drugs and equipment both in the community and the facility. The state has also created FP kits to distribute to newly-weds in rural areas, put up free condom boxes in strategic locations including hospitals, and raised financial incentives for sterilization procedures [17]. SIFPSA (State Innovations in Family Planning Services Project Agency), a joint venture of Government of India, USAID and Government of Uttar Pradesh has designed and implemented several mid- and mass-media campaigns to accelerate the government's efforts for demand generation for family planning in the state [18]. Examples of these initiatives include a multi-media communication campaign called Aao Batein Karein with a primary focus on couples in the age group 17–25 currently not using contraception, Sarthi Sandeshwahini (a mobile video van for promotion of family planning in selected districts) and Sehat Sandesh Wahini (mobile video van project on RMNCH+A campaign). Despite these efforts, Uttar Pradesh continues to face low FP uptake with only 31.7% uptake of modern contraceptive in 2016 [19]. Novel interventions are needed to improve reproductive health and family planning efforts in Uttar Pradesh with particular emphasis on the unique needs of young adults, which make a significant percentage of the state's population. The government of Uttar Pradesh has received support from the a technical support unit (TSU) embedded in the Government of Uttar Pradesh tasked with supporting the state government to increase the efficiency, effectiveness and equity of the delivery of key RMNCH services including family planning. Advancing toward FP2020 goals requires new, systematic research to understand people's decisions to use modern contraceptive methods and the factors that influence them. In this study, using a mixed method approach, we aim to systematically capture the drivers of family planning uptake in Uttar Pradesh from the perspective of households to generate new insights around adoption of modern contraceptive methods to better inform future FP interventions.

## Methods

The study relies on multiple data sets. Ethics approval for the decision games was obtained from the Sigma Research and Consulting Institutional Review Board, New Delhi, India (IRB

number 10025/IRB/D/16-17). Ethics approval for journey mapping was obtained from the Sigma Research and Consulting Institutional Review Board, New Delhi, India (IRB number 10016/IRB/D/16-17). Ethics approval for DLFPS was obtained from the Sigma Research and Consulting Institutional Review Board, New Delhi, India (IRB number 10015/IRB/15-16). Only individuals who consented to participate in the study were included. Written consent to participate in the study was obtained from all participants prior to starting the interview.

We undertook a mixed method approach generating insights from leveraging existing quantitative studies and undertaking new qualitative studies. We used three data sources for our quantitative analyses: two ongoing multi-round government funded surveys, the National Family Health Survey (NFHS) and Annual Health Survey (AHS), and a FP-specific quantitative survey, the District Level FP Survey (DLFPS) [20–24]. The DLFPS collected more extensive information on family planning than the NFHS and AHS including specific information on access to contraceptives and intentions to use methods. As such, it was used to report levels of geographic variability in method mix, analyze the cascade of awareness, intention and use of modern methods, and to identify factors that are associated with modern contraception uptake, by method. The NFHS and AHS were used to determine trends in the use of contraceptives by method. Qualitative methods were designed to generate insights on household decision making, influencers, and internal drivers of FP uptake. Qualitative methods used include in-depth immersion interviews, journey mapping, and decision games.

## Categorizing FP methods

We classified contraceptives in three ways after synthesizing the categories of contraceptives provided in the DLFPS and the NFHS and FP2020 resources. 'Modern methods' included female and male sterilization, intrauterine contraceptive devices (IUCD), injectables, emergency contraception (ECP), oral contraceptive pills (OCP), condoms, the standard days method, and the lactational amenorrhea method (LAM). Temporary effective modern contraception includes IUCDs, injectables, ECPs, and OCPs. Condoms, an otherwise effective and crucial family planning method, were omitted from the temporary effective category because our data showed that its use tends to be haphazard and inconsistent, and therefore not effective. Traditional methods include, but are not limited to, the calendar or rhythm method and withdrawal.

## Analysis approach

Mapping the drivers of behavior enables the systematic categorization of behavioral influences, which can then point towards appropriate interventions to change behavior. To structure our analysis, we holistically considered the potential drivers of behavior using the CUBES framework [25]. According to CUBES, contextual and perceptual drivers combine to act as enablers and barriers along an individual's path from knowledge–encompassing awareness and skills–to intention (or motivation to act) and action and beyond. Layers of influencers can affect these drivers and reach an individual through various channels. Contextual drivers, include factors such as infrastructure (e.g. distance to clinics or availability of supplies), laws, systems and processes (such as referral processes between healthcare providers), demographics, and social norms (e.g. pressure to have many children) [26]. Perceptual drivers include factors such as beliefs around perceived health risks, self-efficacy ('Will I be able to control or achieve an outcome?'), and outcome expectations ('Will the outcome be beneficial or not?') [27]. Emotions, personality, and–largely unconscious and innate–biases are additional perceptual drivers of behavior [28–30]. Influencers can shape a person's drivers through various channels, whether in person or through media. Each of these drivers can make it more, or less, likely that

a person takes action, and together they sum up to an 'action tendency'. The quantitative data provided insights into the stages of change as well as contextual drivers, while the qualitative data offered added information on contextual and perceptual drivers.

## Quantitative methods

We leveraged the three quantitative population-based survey data sets on FP in UP mentioned above. For our analyses we used the latest two rounds of the NFHS: NFHS-3 and NFHS-4, conducted in 2006 and 2016 respectively and the three rounds of the AHS conducted in 2011, 2012, and 2013 [20–22]. The third quantitative data set used was the 2016 District Level FP Survey (DLFPS) conducted by the Technical Support Unit (TSU) for the Government of Uttar Pradesh in 25 high priority districts of UP. Combined the NFHS and AHS, were used to analyze FP method mix in UP overtime. The DLFPS data was also used to analyze geographic variability, the knowledge-intention-action gap in FP method usage and factors associated with FP method uptake. Table 1 shows the quantitative surveys used and their sample sizes.

**Sampling approaches.**   While the sampling methods for the NFHS and AHS have been previously published, this is the first publication of the DLFPS data [19,22]. The DLFPS was administered in 25 high priority districts (HPDs) in Uttar Pradesh. High priority districts were determined by the government based on indicators of the Maternal Mortality Ratio (MMR), % of safe deliveries, Infant Mortality Rate (IMR), % of children 12–23 months fully immunized, Total Fertility Rate (TFR), and Contraceptive Prevalence Rate (CPR)–Modern Method [31]. Its sample size was determined based on prevalence of modern contraceptive users in the AHS 2012–2013. The survey used a two-stage cluster sampling design. First five blocks were selected in each district in proportion to population size. Second, frontline worker (ASHA) areas were used as primary sampling units (PSUs) for random selection. An ASHA area refers to a population catchment area of approximately 1000 people, who the ASHA caters to for promoting good health practices and behaviours. The number of PSUs in each district was decided considering a total of five interviews could be conducted. After random selection, five women were interviewed in each PSU by a female investigator between April 2016 and August 2016.

**Table 1. Quantitative survey Uttar Pradesh sample sizes.**

| Survey | Years | UP Sample Size |
|---|---|---|
| **National Family Health Survey (NFHS)** | 2005–2006 | n = 10,026 households (12,183 women, and 11,458 men) |
| | | Statewide |
| | 2015–2016 | n = 76,233 households (97,661 women, and 12,939 men) |
| | | Statewide |
| **Annual Health Survey (AHS)** | 2010–2011 | n = 847,297 households (4,528,409 population), 797,508 ever married women) |
| | | Statewide |
| | 2011–2012 | n = 869,959 households (4,750,285 population), 832,614 ever married women) |
| | | Statewide |
| | | Statewide |
| | 2012–2013 | n = 883,613 households (4,808,503 population), 832,614 ever married women) |
| | | Statewide |
| | | Statewide |
| **District Level Family Planning Survey (DLFPS)** | 2016 | n = 13,182 married women |
| | | 25 High Priority Districts |

**Statistical analysis.**   Descriptive statistics of the DLFPS survey data were used to calculate the FP awareness-intention-usage cascade, the FP awareness-intention-action cascade by limiting and spacing methods, and the district-wise method mix. Trends in method use over time were calculated by collation of reported values from the NFHS and AHS.

In the FP cascade, to identify the predictors of intention to use of modern contraceptive methods, separate multivariate logistic regression models were run on the DLFPS survey data for the following dependent variables: had intention to use female sterilization, had intention to use pills, had intention to use condoms, and had intention to use IUCD. For independent variables, a wide range of variables (socio-demographics, frontline worker counselling, access, media influence, awareness, frequency of sex, etc.) with potential to influence modern contraceptive method uptake were considered, after controlling for multicollinearity. Further, to understand the drop off from intention to using a particular method to using that method, chi-square test of independence was done to identify differences in various contextual factors between individuals that were currently using a method versus those had the intent to use a method but were not currently using it.

## Qualitative methods

In order to probe deeper into the additional drivers and decision dynamics, the quantitative analysis was supplemented by de novo qualitative research on FP in UP using three methods: in-depth immersion interviews, journey mapping, and decision games. These studies around FP were conducted as part of a larger study of RMNCH behaviors. As a first step, in-depth immersion interviews were conducted to provide preliminary insights into factors influencing RMNCH behaviours, which were consequently used to inform the design of journey mapping and decision games. We conducted research on households, other household influencers, frontline workers, and facility staff involved in FP counseling and supply. For this paper, we analyzed the data from households and their influencers only. All qualitative data collection occurred in 2016.

**Sampling.**   The sampling process for the in-depth immersion interviews consisted of two stages: selecting the districts and blocks from which households were drawn and selecting households. First, districts were selected based on being within a distance of 30 to 130 kilometers from the state capital, Lucknow, as well as their designation as a high priority district from the government. Within each district, blocks were chosen to reflect a combination of the following characteristics: whether blocks had a government high-priority program in place or not, and whether community literacy levels, the percentage of rural dwellers, and the percentage of community members coming from Scheduled Castes/Scheduled Tribes were high or low (see S1 Appendix for block selection methodology) [22]. 10 different blocks in 4 districts were selected of which 5 were high priority and 5 were not.

For journey mapping and decision games blocks in rural Uttar Pradesh were selected based on the percentages of currently pregnant women registered, women who received 3 antennal check-ups, women who delivered in a health facility, women who received any postnatal care home visits after delivery, couples currently using any modern contraceptive method, and representing self-help group (SHG) and non-SHG. The 5 indicators were classified as good, average and poor based on cut offs derived from the distribution of the data (see S2 Appendix for details on datasets used and selection criteria applied). An overall block classification into good, average and poor was done based on the block's overall performance across these indicators. For journey mapping, a total of 6 blocks (3 TSU, 3 non-TSU) from 6 different district were chosen and for decision games 24 blocks (18 TSU, 6 non-TSU). For journeys, 6 families-3 that had never used a modern FP method and 3 that did use a modern method were selected.

The chosen families were selected to exclude currently pregnant women or those who had recently experienced childbirth. The chosen families represented a mix of sociodemographic characteristics such as religion, sex of last-born child, caste, economic status and woman's age. For decision games, participants were selected to ensure uniform caste, age, SEC, and education in each group. They also selected for a balance in stage in the RMNCH journey (from first trimester pregnancy to 6-month-old newborn), place of delivery, if they had a death of a child before 6 months, number of pregnancies, use of family planning, and religion. Participants were recruited across 24 blocks across 21 districts in UP. Table 2 shows the sample size for each method.

**Household in-depth immersion interviews.** Literature reviews and stakeholder discussions were conducted to design the content of the in-depth immersion interviews, which revealed several factors that could influence family planning behaviors. These factors were structured using the CUBES framework to design the interview discussion guides [25]. The discussion guides (S3 Appendix) included sections on intentions, actions, influencers and beliefs. In-depth immersion interviews of household members provided preliminary qualitative insights into the factors driving FP uptake in UP. In-depth Immersion interviews were approximately 90-minute-long, face to face, group, semi-structured interviews where households were asked a series of open-ended questions about their experiences and beliefs around FP [19]. Interviews were conducted in 9 blocks across 4 districts in UP. We interviewed those household members hypothesized to be primary influencers on FP. In addition to women and men, this included mothers-in-law (MIL) and sisters-in law (SIL). Women were interviewed separately from influencers (MIL, husband, and SIL) to allow them to respond independently. The interviews were analyzed using thematic analysis [32]. Themes included infrastructure, decision dynamics, social norms, past and current behaviors, beliefs, emotions, and stages of change. Findings from in-depth immersion interviews were then used to inform the design of the journey mapping and decision games.

**Journey mapping.** Journey mapping is a market research approach that systematically tracks people's experiences and interactions with a product or practice over time. The method also captures the beliefs that a person forms about the practice, as well as key influencers, that affect decisions to engage in or avoid the product or practice [33]. The process of journey mapping better homes in on an individual's perspective on a specific issue over time and allows for matching multiple factors influencing behavior and beliefs to respective stages of change, while also taking the perspective of other influencing individuals into account. The journey mapping interviews in this study mapped beliefs around FP from awareness to intention and habit formation. Members and influencers of 6 households completed journey mapping for FP including: women, husbands, mothers-in-law, and "other influencers" (friend or family member). Interviews were conducted using discussion guides (S4 Appendix) and journey mapping kits consisting of belief cards, pre-populated sticker libraries, and journey books.

**Decision games.** Ethnolabs are an audio-visual decision game developed by the behavior change company Final Mile. Decision games are designed to overcome the biases such as fear

**Table 2. Qualitative method sample sizes.**

| Method | Years | Sample Size |
|---|---|---|
| In-depth Immersion Interviews | 2016 | n = 35, 14 women, 8 husbands, 10 mothers-in-law (n = 10), and 3 sisters-in-law |
| Journey Mapping | 2016 | women (n = 6), husbands (n = 6), mothers-in-law (n = 6), and "other influencers" (friend or family member) (n = 6). |
| Decision Games | 2016 | n = 61 women, 60 men, 55 mother-in-law (MIL) |

of value judgment, social desirability, and expectation of higher self-control in future that may make respondents in research-settings claim behaviors that may be not be true in real life [34]. The games simulate the real-world context of participants, observes their hypothetical decision-making, and relates their choices to underlying drivers of behavior. Because FP is a topic often considered private, discussing it in the hypothetical game context allowed for participants to disclose beliefs about the topic they might not in personal interviews. 5 participants with similar profiles in terms of age, gender, and income played the 90-minute game at a time. During the game, participants were given live or audio-visual descriptions of unique situations that end with a decision conundrum, with a set of 3 different scenarios (S5 Appendix) among which the participants must choose. Participants chose based on what they believe the majority in the group will select as the most likely outcome among the options, rather than having to state their own opinion, which is designed to help circumvent the typical self-report biases mentioned above. The game involved a scoring system to motivate players. Afterwards, participants engaged in discussions about their experience, decisions, and preferences in the game (see S6 Appendix for discussion guides). Each scenario could test for the involvement of several likely behavioral focus areas. Three FP scenarios were embedded in a larger game of 10 scenarios related to an array of reproductive and maternal health behavior. See Table 3 for the drivers of FP assessed in the decision games.

## Results

### Minimal change in FP usage and types of contraceptives used over time

Combined analysis of the NFHS and AHS reveals that the FP method mix in UP has changed little from 2006–2016 (Fig 1). Overall use of modern contraceptives (mCPR) in UP, however, has increased slowly since 2006 from 29.3% to 31.7% in 2016. This trend of little change in mCPR is true for India overall as well, with mCPR decreasing slightly to 47.8% in 2015–16 from 48.5% in 2005–06 [16]. Despite consistency in method mix, the DLFPS data shows regional variability among high priority districts in UP in method use (S7 Appendix). The mCPR in Mirzapur is 55%, but it is only 13% in Balrampur. This finding suggests understanding the complex range of factors that influence uptake is very important.

### Awareness-Intention-Usage gaps in FP

To analyze the reasons for the lack of change in mCPR and the types of methods used, it was first necessary to determine where along the action tendency path towards contraceptive use women tend to be. The DLFPS data showed that overall general awareness of modern contraceptives, measured as whether the woman had heard of the method, is high among women in UP. 99% of women are aware of any modern method and 94% are aware of any effective

**Table 3. Drivers of FP assessed in decision games.**

| FP Scenario Focus | Hypotheses |
|---|---|
| Key reasons for using FP | 1. Good for woman's health<br>2. Good baby's health<br>3. Good for financial reasons |
| Key reasons for resistance to FP | 1. Fertility is god's will, we have no control<br>2. Fear of side-effects<br>3. Woman has low agency |
| Key reasons to use of specific contraceptives | 1. OCP for control over fertility<br>2. Condoms so husband can make the decision<br>3. IUCD for long-term certainty |

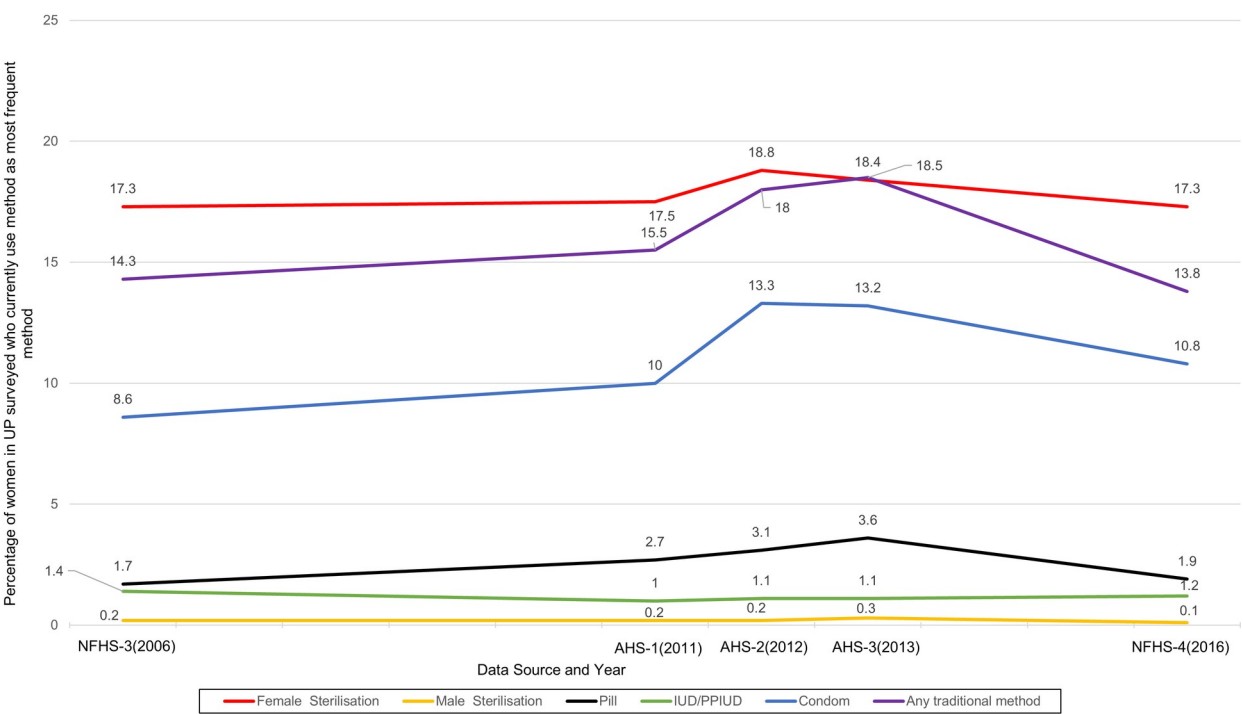

**Fig 1. Little change in distribution of contraceptive use by method overtime in UP between 2006 and 2016.** Figure shows combined analysis of responses in quantitative surveys described in methods (NFHS and AHS) to the question: "Which method are you currently using to delay pregnancy?", considering the highest/most used method. Lactational amenorrhea method (LAM) has been deducted from the "Any traditional method" indicator in AHS to match the corresponding NFHS definition for this indicator.

temporary method. Awareness of specific methods ranges from 97% for female sterilization to 84% for IUCD and Injectables. Awareness, however, does not measure the person's understanding of the side effects of an individual method. The two exceptions are awareness for emergency contraception (ECP) and lactational amenorrhea method (LAM) which are lowest at 29% and 19% respectively. In addition to high awareness, there are gaps between past use and current use of family planning methods along the cascade. 76%, 49% and 15% of women have ever used any method, modern methods, and effective temporary methods respectively, while only 65%, 31%, and 4% of women are currently using methods in the three categories, as shown in Fig 2. Different factors can influence discontinuation. The three main reasons reported for discontinuation of a method are: infrequent sex (28%), method failure (23%) and wanted to become pregnant (29%). Combined, the drop off between rates of awareness and use as well as between past use and current use demonstrate discontinuity in contraceptive use that requires further investigation.

Fig 2 shows the entire cascade of method use, from wanting to space or limit childbearing to use of effective temporary modern methods. Desire to space is defined as women who do not want a child now/soon and desire to limit includes those that do not ever want a child and those already sterilized. Desire to space or limit childbearing is also high at 86%. Intent to use is defined as women already using FP methods or non-users who answered they want to use FP in the future. Despite the high percentage of women who want to limit or space, only 65% intend to use any FP method, and only 53% use any method. This suggests that awareness and need alone are necessary, but often insufficient in building intention to use family planning. Moreover, only 40% of women intend to use any modern method and only 31% use any modern method. The additional drop-off between intention and action again suggests that

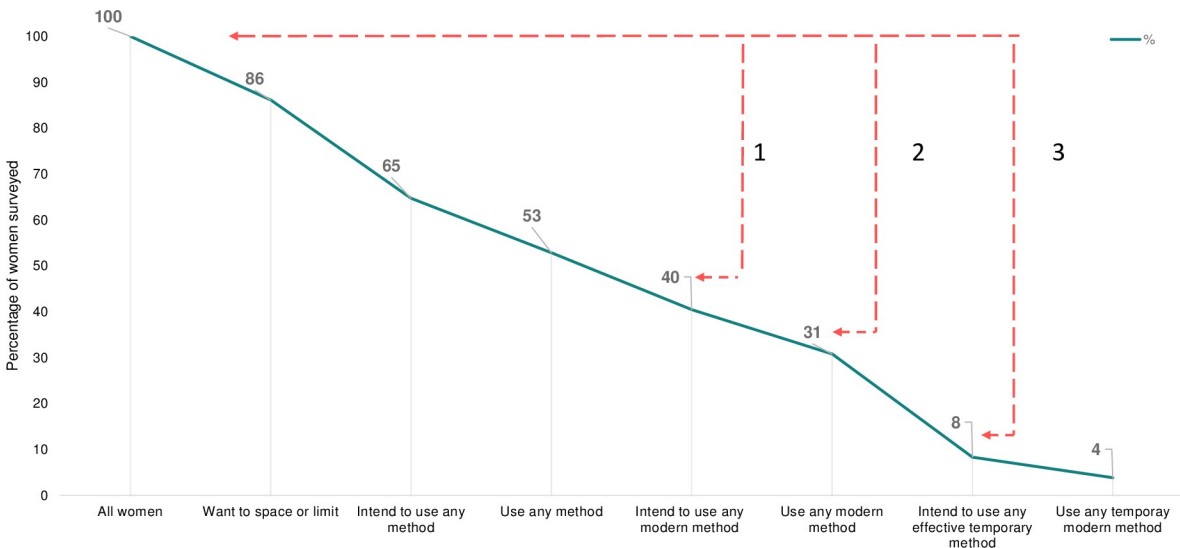

**Fig 2. Family planning awareness-intention-use cascade.** The figure shows descriptive analysis of 2016 DLFPS (n = 13,182 married women ages 15–49). Despite high awareness of modern methods (99% of women are aware of modern contraceptives and 94% are aware of modern temporary ones), there are large gaps between awareness, intention, and use. As shown, there is a 46% drop (bracket 1) between women who want to space/limit and women who to intend to use a modern method, a 55% drop (bracket 2) between wanting to space/limit and using any modern method, and a 78% drop (bracket 3) between wanting to space/limit and intending to use any modern temporary method. "Want to space" is defined as the response "not wanting to have child now/soon". Any modern method includes female/male sterilization, IUCD, OCP, ECP, injectable, condom, standard days method, and LAM (note that % of standard days method and LAM are minimal). Any modern temporary method includes IUCD, ECP, OCP, and injectable. Only non-users were asked about intention to use methods in the DLFPS, hence, added current users to group because we assume that current users had intent.

intention may also at times be insufficient to cause uptake. Overall, the data shows a 46% drop from wanting to limit or space childbearing to intending to use modern methods to do so and an additional 11% drop in those who ultimately do. Understanding why these drop-offs exist will help to design effective interventions to drive uptake. Only 8% of women intend to use a temporary modern method and only 4% do so.

S8 Appendix shows that the cascade differs for women who want to limit pregnancy (65% of sample) versus those who want to space pregnancies (21% of sample). Intent and use are higher for women who want to limit pregnancy than for those who want to space. Among women who wanted to limit, 79% intended to use any method and 65% used any method, while among women who wanted to space pregnancies only 51% intended to use any method and only 39% used any method. These findings reveal that the awareness-intention gap surrounding modern contraceptive methods is a significant issue contributing to FP uptake in UP. Results indicate intention to use is limited by factors beyond awareness. It also indicates while raising awareness of methods choice is necessary, it is not sufficient to build intention to use any of those methods.

## Influence of contextual factors varies by modern method

There are a number of other contextual factors that may drive intention to use a modern contraceptive method. Four multivariate logit regression models were run on the DLFPS survey data to determine contextual predictors of intention to use female sterilization, condoms, IUCDs, and pills. Contextual drivers included in the model were demographics such as age and caste, reproductive history, contraceptive access, and message exposure. Ultimately, the influence of contextual factors varied by method. Older women were more likely to intend to opt for female sterilization, but younger women were more likely to intend to opt for

condoms. Muslims were less likely to intend to opt for permanent methods like sterilization, unlike Hindus, but more likely to intend to opt for condoms and pills. Woman's and husband's literacy in intention to use of different FP methods only mattered for condoms. Scheduled Caste/Tribe populations were more likely to have an intention to get sterilized but were less likely to have an intention to use IUCDs. Economic status was positively associated with the intention to use sterilization and pills. Those that were not aware of other methods beyond female sterilization and condoms were more likely to have an intention to use female sterilization and condoms, respectively. This suggests that some women's choice of these two methods is largely driven by the lack of awareness of other available methods. Having 1 or more son was a significant positive driver of all but IUCD intent to use.

Women that perceived accessing sterilization and IUCD to be hard, respectively, were less likely to intend to use the method, whereas those that perceived accessing other modern methods to be hard, other than sterilization and IUCD, respectively, were more likely to intend to use the method. Receiving counselling on the method by frontline workers (FLWs) and facility staff increases the likelihood of IUCD, pill and condom (FLWs only) intention to use, but not for sterilization intention to use. These findings suggest that the influence of contextual factors vary by modern method type. S9 Appendix shows the complete results of the four regression models. These findings suggest that different methods are preferred by individuals varying in their demographic characteristics, awareness levels, perceptions of access and therefore careful targeting by method should be considered.

Next, to understand the intention to action segment in the FP cascade, we looked at the differences in contextual factors between women who were currently using the method (referred to as 'users' for simplicity hereafter) and women who had the intention to use the method but were not using it (referred to as 'intenders' for simplicity hereafter). Across methods, intenders had a significantly higher proportion of younger women, women that did not know whether it was easy or difficult to access the method and women that had infrequent sex, as compared to users. Intenders of all methods, except IUCDs, had significantly higher proportion of women with fewer number of sons than their corresponding users. Sterilization and pill intenders had a significantly higher proportion of poor women as compared to sterilization and pill users. IUCD intenders had a significantly higher proportion of Muslims as compared to IUCD users. S10 Appendix shows the complete results. These findings suggest that certain contextual factors need to be specifically addressed to close the intention to action gap, and that these can be different for different methods.

## Household decision making varies by method

Personal relationships within households are another important driver of FP behavior and household dynamics play a role in decision-making. Decisions around family planning are a product of discussions and beliefs shared between husbands and wives. Data from interviews, journey mapping, and decision games revealed different household decision making dynamics by modern contraceptive method. For IUCDs and sterilization, there was evidence of a syncretic dynamic, where the woman and man decide together. During the decision games when a group of men was asked who made decisions about sterilization, one responded, "*both [husband and wife] should decide.*" Condoms, in contrast, may be decided by men, and the woman may have almost no power over this decision. One woman during the decision games explained, "*we only use condom, my husband don't want to go for any other method.*" Women are more autonomous, however, when it comes to injectables. One woman preferred injections because "*there is no risk of any disease in case of injection, in Copper T there is more risk therefore I find injection to be proper.*" Overall, wives have little exposure to FP outside the family,

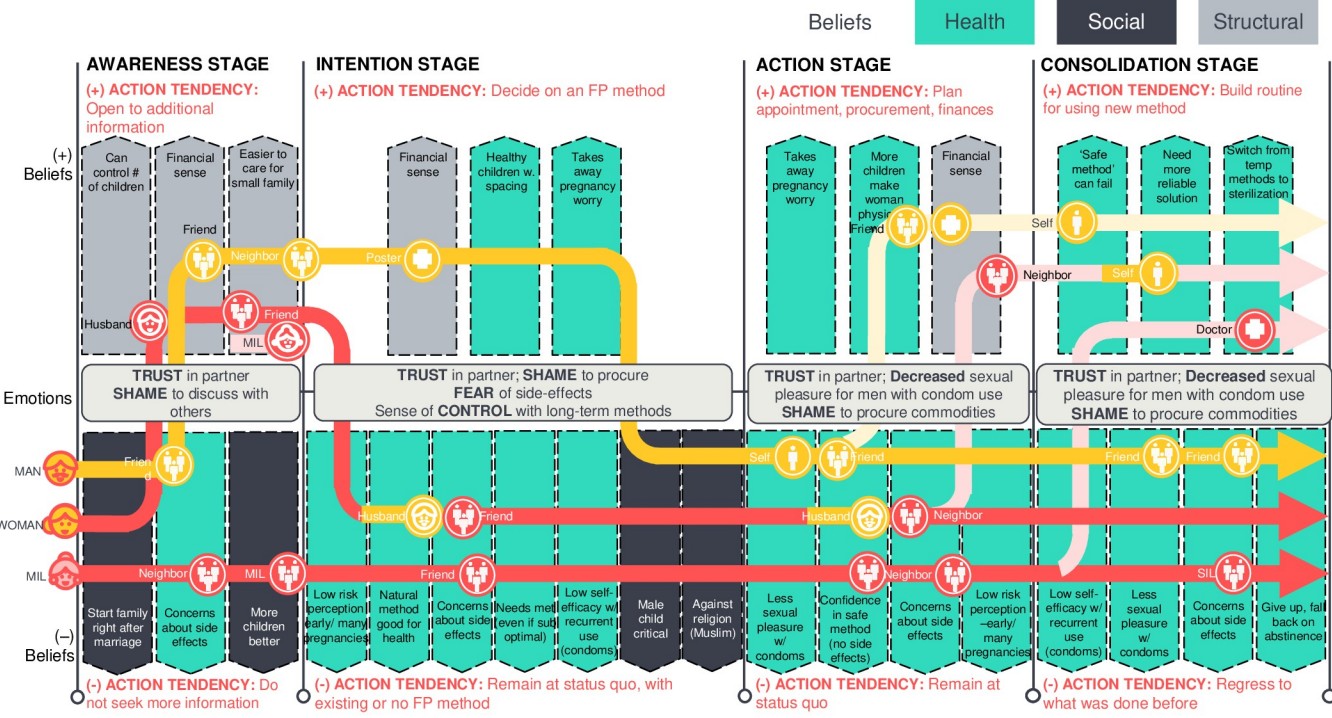

**Fig 3. FP journey.** Figure shows a summary of qualitative journey mapping outputs (n = 6 women, 6 husbands, 6 mother-in-law (MIL), 6 ASHA, and 6 friends/family). The combined journey mapping analysis illustrates the different beliefs that either facilitate (+) or inhibit (-) progression to the next action tendency towards contraceptive use. Health beliefs are those about health risks to the mother and/or baby. Social beliefs include community norms and structural beliefs include anything related to family structure. At each action stage (awareness, intention, action, and consolidation) health, social, and structural beliefs determine whether a woman, husband, or mother-in-law (yellow and red lines) will seek more information, intend to use FP, use FP, and form FP habits. Influencers (circles) contribute to stakeholders' beliefs at each stage.

and are mostly influenced by their husbands. One woman explained during an in-depth immersion interview that it was her husband alone who chose to use condoms, a method which she had no awareness of until marriage. Men take part in more discussions with friends within the village and are exposed to outside media, and MILs have a neighbor network in the village.

### Risk perception, shame, and fear often drive FP decision making

In addition to household dynamics, a number of perceptual behavioral drivers can influence FP behaviors. Shame, social pressures, and fear of health risks were all found to play a role in decision-making. Fig 3 shows the interaction role of these drivers across the action pathway towards family planning uptake for different household members.

### Strong social norms against and shame to obtain FP are widespread

The qualitative data shows that shame, often in relation to violating social and religious norms, is a key perceptual barrier to contraceptive use in UP. Analysis of journey mapping and decision games shows that social norms influence intentions and uptake of modern methods in UP. Strong norms to demonstrate fertility, have many children, and have at least one male child were all cited in interviews as sources of reluctance to use methods. As one husband explained during the decision games, *"I have four children. At the start I had one boy. But I*

*wished to have two more boys. Due to this I now have two girls and one boy. So now I will have four children.*" One woman shared her perspective on social norms: "*Everyone who got married at the same time as me are having two kids and I did not even have one so people used to say everyone is having kid why are you not having a kid yet. So, I thought I should also have a kid now.*" In addition to these strong norms, general feelings of shame to procure contraceptives by women and men were also mentioned as barriers to uptake. On the topic of obtaining information about contraceptives, one woman admitted, *"I will ask my sister in law because I am too shy to ask anyone else."* For some individuals, religious norms against FP played a role as well.

### Risk perception is high for contraceptive side-effects, but low for early or unspaced pregnancy

Another important perceptual driver is risk perception, which was found to be high for some modern contraceptive methods. Risk was perceived both around potential side effects of methods, ineffectiveness and in the case of condom use, decreased sexual pleasure. It is especially high for more intrusive and surgical methods: IUCDs and sterilization. Risk perception around infertility was notably high for temporary methods, with women suggesting that contraceptives could cause them to become infertile. "By using Copper T women lose more blood and womb is damaged." This misconception is mentioned for many modern temporary methods and appears to be a significant barrier to use. One woman commented on the IUCD during an in-depth immersion interview: *"Even if someone kills me, I will not get it done, I don't like Copper T and all that, due to bleeding, there is a risk factor."* Risks were also noted for OCPs. A husband commented, *"My wife is weak and I don't think pills are safe it does not suit anyone."*

In contrast, risk perception around the consequences of not using any FP method appeared at times to be absent or low, such as little perceived risk around early pregnancy, multiple pregnancies, or low spacing. When asked about how long she waited to have children after marriage, a woman responded, "*We did not think about it like that, when it happens it happens.*" This suggests that the causal links are not made between FP and the risk of these undesired outcomes. Households often do not always recognize the potential financial or health risks to the mother and child posed by unplanned pregnancies, having children closer together than desired or more children that desired. For those that use FP, however, financial risk of unplanned pregnancy can be key rational. One woman commented on her husband's beliefs around FP, "*He said it's good that we are going to have baby, big family is not affordable but let our first child come.*"

### Decision making pathway

The above sections highlight several contextual and perceptual drivers of FP behaviors that, when considered in combination, indicate a current decision pathway for families has three potential main branches (Fig 4). The pathways ultimately lead to either adopting a permanent method or giving up on FP altogether. First, families may avoid FP, stemming from a combination of strong social norms against FP, low risk perception around pregnancy, concerns about side effects, and feeling ashamed to discuss FP. Second, families may adopt temporary modern methods as they want to control family size and better health and consider finances. After adopting temporary contraceptives, some families may eventually give up on FP after experiencing difficulty and a low sense of control with temporary methods, a belief that abstinence is safe, feeling ashamed to procure contraceptives, and experiencing decreased sexual pleasure with some methods. Alternatively, families finally adopt a permanent method after

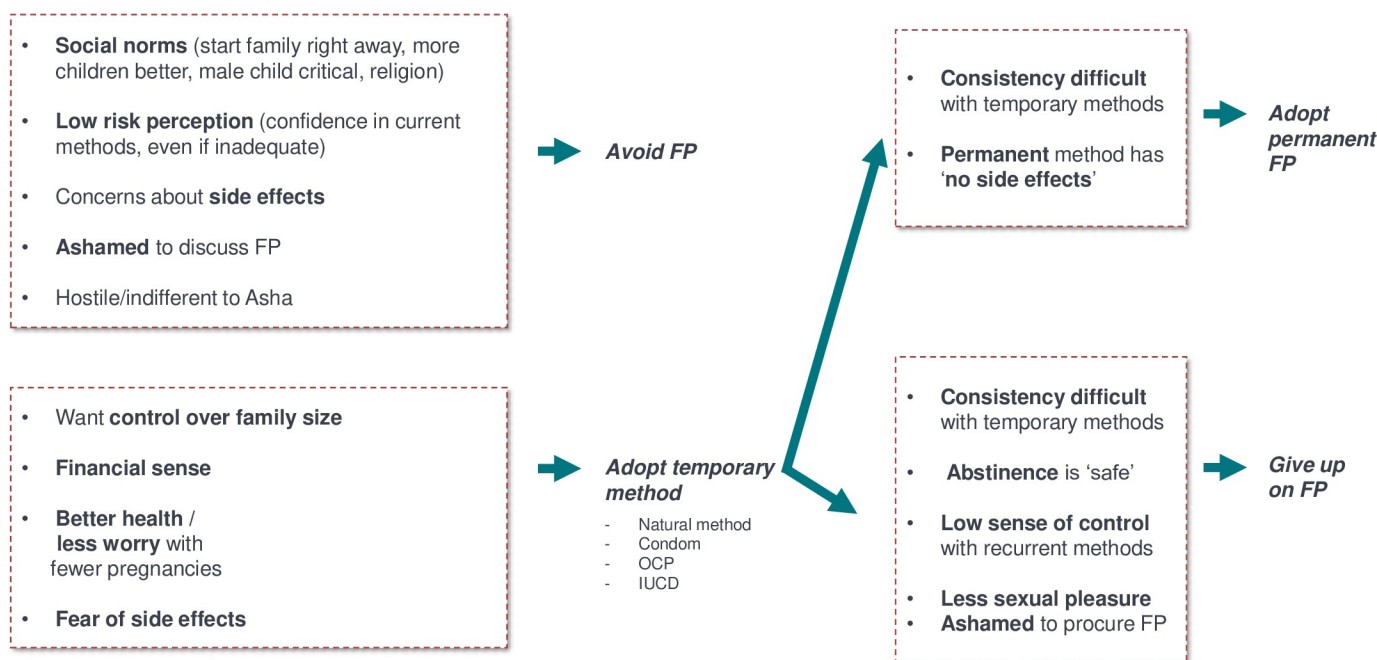

**Fig 4. FP decision tree.** Analysis of qualitative findings suggest three potential pathways for FP. Interaction of social norms, risk perception, beliefs about side effects, shame, and other beliefs lead families to 1. Avoid FP altogether, or 2. Adopt a temporary method and eventually a permanent one or 3. Adopt a temporary method but additionally give on FP altogether. Additional quantitative research is needed to confirm pathways.

having difficulty using temporary methods consistently, and because of a belief that permanent methods have no side effects.

## Discussion and conclusion

In this study, we leveraged both existing quantitative data sets and a combination of novel qualitative methods to explore a comprehensive set of drivers of family planning use in Uttar Pradesh, India. The quantitative data showed little change in both overall use of modern contraceptives and the types of FP methods used between 2006–2016. To understand the reasons for the lack of change, we first examined where along the journey to action women were. We found that while both awareness of modern contraceptives and desire to space or limit childbearing was high, intent to use them was low (and especially low for temporary methods). There was an additional drop off from intention to actual use. To uncover the sources of this low intention to use, we first looked at factors measured in the quantitative data such as access, age, education and other demographic variables. While these factors partially predicted intention to use, their influence varied by method. We also identified some specific contextual factors that explained the intention to action gap. The qualitative data from interviews, journey mapping, and decision games provided more nuanced insights on the individual reasons for low intentions and uptake not captured in the quantitative data. First, it revealed that different beliefs and emotions played a large role in shaping uptake. Social norms against and shame to obtain modern contraceptives was widespread, while risk perception was high around side effects of methods–particularly fear of infertility from temporary methods, but low around the consequences of not utilizing FP for having an unplanned additional or inadequately spaced child. Combined these findings suggest three potential decision-making pathways for how different households approach the FP pathway: those that disregard it all together, those that

adopt temporary methods but eventually give up, or those that adopt temporary methods and eventually turn to permanent methods. Previous data has shown that another pathway could exist where households directly adopt permanent methods without having used any other contraceptive methods [23].

In addition to its expansive use of data, this study is, to our knowledge, the first to conduct a systematic assessment of FP behavioral drivers, both perceptual and contextual, and assess the influence of all stakeholders involved in the FP landscape (women, husbands, frontline workers, and mothers-in-law). These approaches allowed for a comprehensive exploration of FP drivers in Uttar Pradesh that provides insights to inform interventions. One key insight is the existence of a pronounced awareness-intention-action gap. The finding stresses the need to shift focus towards building intention, in addition to ensuring access. This research also reveals that in order to build intention to use modern contraceptives, interventions should also work towards changing perceptual drivers such as fear of infertility and side-effects, which is largely prevalent in the communities. Current interventions, including family planning counselling services in communities and facilities need to be strengthened to address these gaps. In addition, focus should be placed on engaging men and framing contraceptive use through economic arguments, as opposed to solely health-related justifications [35–37].

Building intention will take time and cannot be achieved with quick interventions. There are, however, several ways to begin the process. Given the prevalence of fear of side-effects and infertility, risk perception around certain methods should be reduced and risk perception around unplanned pregnancies should be heightened. Women's fear of side effects should be mitigated by addressing misconceptions (such as loss of fertility), providing strategies to mitigate real side effects, and helping women better understand their menstrual cycle. Messages should be targeted to fit the key barrier of specific methods. For example, fears of infertility caused by OCPs should be addressed differently than fears of bleeding caused by IUDs. Risk perception around unplanned pregnancies and other consequences of not practicing FP, including inconsistent and haphazard use, should be amplified, ideally by using narrative examples bringing risk into the familiar world and emphasizing modern contraceptives as the means of combatting said risk. Interventions should explicitly address contraceptive effectiveness of different methods to provide men and women with accurate and actionable information [38,39]. Knowing which methods are more or less effective than others in preventing pregnancy can help individuals make more educated decisions on their contraceptive choices.

Second, the feeling of shame to discuss and procure contraceptives should be resolved for men and women by normalizing the topic (e.g. by leveraging key opinion leaders, inserting FP messages into pop culture, and training women in the skills to use contraception) [40,41]. Interventions designed around shame could also provide information or purchasing opportunities anonymously.

Finally, we agree with the growing body of research suggesting men are key decision makers and should not be an afterthought for intervention [17]. Their barriers and channels through which they can be reached are not always the same as their wives'. For example, highlighting the consequences of financial risk on family health and wellbeing could be a key message to leverage for men.

In addition to the suggested interventions for building intention, because the drivers and barriers to contraceptive use are varied by person and method, FP offerings should adopt a private-sector approach to product launch and uptake [42]. This includes segmenting women and couples on their psycho-behavioral differences in barriers to contraception use, and a market-segmented approach to stimulating uptake. Products can be targeted to different consumers based on the different barriers and concerns users face, in addition to medical considerations. This should be supplemented with interventions that work on shifting community

norms and attitudes. Given noted geographic heterogeneity in method use, programs should also target based on geographies.

The findings presented here also have limitations. The quantitative regression models' results are associative and not causal. Perceptual factors that emerged relevant for method use in the qualitative findings were not measured and included in the models. The relatively small sample size of the qualitative research, and the use of quantitative data from only 25 high priority districts within one state, may limit the generalizability of the findings to other settings. Additionally, because of the exploratory nature of the qualitative research, the relative importance of each driver and their interactions cannot be quantitatively determined. This calls for future quantitative research building on the insights presented here to close this gap. Future studies should also address remaining questions, such as how different decision dynamics vary by method and what characterizes the women who independent decision-makers are. Research should also explore what concerns about side effects specifically consist of, who influences them, and how they can be addressed. Additionally, more should be learned on what differentiates FP behaviors between women and men who want to space versus limit childbearing.

Increased use of modern contraceptive methods in low- and middle-income countries is an essential step towards reaching the Sustainable Development Goals and improving health, educational, and economic outcomes for households. This study offers a comprehensive look into the drivers of contraceptive use for different household members in one geography. Our findings suggest that interventions must focus on increasing intention, rather than awareness, to use modern temporary methods by focusing on shifting key perceptual drivers of behavior.

## Supporting information

**S1 Appendix. Block selection for in-depth immersion interviews.**
(XLSX)

**S2 Appendix. Datasets and methodology for selecting blocks for journey mapping and decision games.**
(DOCX)

**S3 Appendix. In-depth immersion interview discussion guide.**
(PDF)

**S4 Appendix. Journey mapping discussion guide.**
(PDF)

**S5 Appendix. Decision game scenarios related to FP.**
(DOCX)

**S6 Appendix. Discussion guides for post decision game discussions related to FP.**
(DOCX)

**S7 Appendix. Regional variability in method mix in UP.**
(DOCX)

**S8 Appendix. The cascade for women who want to limit pregnancy versus those who want to space pregnancies.**
(DOCX)

**S9 Appendix. Contextual predictors of current intention to use female sterilization, condoms, IUCDs, and OCPs.**
(DOCX)

**S10 Appendix. Contextual differences between intenders and users.**
(DOCX)

## Acknowledgments

We thank Upstream Thinking, Final Mile and Market Resonance for assistance with the research studies. We also thank Sofia Braunstein and other colleagues at Surgo Foundation for their inputs on the manuscript.

## Author Contributions

**Conceptualization:** Mokshada Jain, B. M. Ramesh, Preeti Anand, Shiva Halli, James Blanchard, Vikas Gothalwal, Sema K. Sgaier.

**Data curation:** Mokshada Jain, B. M. Ramesh.

**Formal analysis:** Mokshada Jain, B. M. Ramesh, Shajy Isac, Elisabeth Engl, Sema K. Sgaier.

**Funding acquisition:** Sema K. Sgaier.

**Investigation:** Mokshada Jain, B. M. Ramesh, Shajy Isac, Elisabeth Engl, Sema K. Sgaier.

**Methodology:** Mokshada Jain, B. M. Ramesh, Shajy Isac, Sema K. Sgaier.

**Project administration:** Mokshada Jain, B. M. Ramesh, Shajy Isac, Hannah Kemp, James Blanchard, Vikas Gothalwal, Sema K. Sgaier.

**Resources:** Sema K. Sgaier.

**Supervision:** Mokshada Jain, B. M. Ramesh, Hannah Kemp, James Blanchard, Vikas Gothalwal, Sema K. Sgaier.

**Validation:** Yael Caplan, B. M. Ramesh, Shajy Isac, Sema K. Sgaier.

**Visualization:** Mokshada Jain, Yael Caplan.

**Writing – original draft:** Mokshada Jain, Yael Caplan.

**Writing – review & editing:** Mokshada Jain, Yael Caplan, B. M. Ramesh, Shajy Isac, Preeti Anand, Elisabeth Engl, Shiva Halli, Hannah Kemp, James Blanchard, Vikas Gothalwal, Vasanthakumar Namasivayam, Pankaj Kumar, Sema K. Sgaier.

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
