## [Decision Letter · Decision Letter 0]

10 Sep 2020

PONE-D-20-21461

Understanding drivers of family planning in rural northern India: an integrated mixed-methods approach

PLOS ONE

Dear Dr. Sgaier,

Thank you for submitting your manuscript to PLOS ONE. After careful consideration, we feel that it has merit but does not fully meet PLOS ONE’s publication criteria as it currently stands. Therefore, we invite you to submit a revised version of the manuscript that addresses the points raised during the review process.

We look forward to receiving your revised manuscript.

Kind regards,

Kannan Navaneetham, PhD

Academic Editor

PLOS ONE

Journal Requirements:

2. Please include a copy of the interview guides used in the study, in both the original language and English, as Supporting Information, or include a citation if it has been published previously.

Reviewers' comments:

Reviewer's Responses to Questions

**Comments to the Author**

1. Is the manuscript technically sound, and do the data support the conclusions?

Reviewer #1: Partly

Reviewer #2: Partly

2. Has the statistical analysis been performed appropriately and rigorously? 

Reviewer #1: I Don't Know

Reviewer #2: No

3. Have the authors made all data underlying the findings in their manuscript fully available?

Reviewer #1: Yes

Reviewer #2: Yes

4. Is the manuscript presented in an intelligible fashion and written in standard English?

Reviewer #1: Yes

Reviewer #2: Yes

5. Review Comments to the Author

Reviewer #1: Title- The title of the paper should mention the state of Uttar Pradesh (instead of saying northern India, since that’s the only state referred to in the paper.

Line 81- Total fertility rate for India is 2.2 (not 2.3)

Para 81-93- This paragraph should bring in the fact that in 2017 the Indian Government introduced three new spacing methods, including injectable contraceptives, in the basket of contraceptive choices as a move to shift emphasis on spacing methods.

Para 81-93- Given that this paper focuses on understanding the complex drivers of family planning uptake, the efforts made by the state of Uttar Pradesh, as described in the abstract, should also include demand generation efforts including any prominent social and behaviour change communication campaigns.

Para 111-119- While condoms have been omitted from the categorization, they are important as they are the only spacing method involving men and their uptake is a critical indicator of male engagement in family planning.

Results

Para 257-263- It must be noted here that while the mCPR in UP hasn’t changed much between 2006-16, comparison of NFHS3 and NFHS4 does not show much change in India’s national mCPR between 2005-06 and 2015-16 either.

Para 327-338- What about female education/employment which would have a role in determining the decision making power of women and their ability to negotiate contraceptives. What is the reference for lines 333-335?

Paragraph 394-415- The prevalence of myths and misconceptions around contraceptives as well as the limited uptake of contraceptives mentioned in previous paragraphs point towards the need to strengthen family planning counselling services which should also be highlighted.

Overall- Though the paper touches upon the complex drivers of family planning in India, the background section needs to be strengthened further to present a stronger case for the family planning landscape in India where women do not have the ability to make their own fertility decisions as these decisions are made by the husband and family on her behalf. The second important point to be noted basis data from the fourth national family health survey is the high unmet need for family planning among women (which is defined as percentage of women in their reproductive age group who want to access contraception but are unable to do so due to various reasons). This is again an important indicator of women’s inability to take control of their fertility decisions. Greater emphasis on family planning counselling services and investments in behaviour change communication strategies are extremely important for demand generation.

The other point that needs to be brought out in the background section is UP’s large young population, who have distinct reproductive health needs which needs to be prioritized too.

Reviewer #2: Thank you for giving me the opportunity to review the article. The article is important and timely. Please find some suggestions. Hope those will be helpful and will strengthen the article for its contribution to future research on this topic.

Specific comments:-

Abstract:

A disconnection between result and conclusion. The results showed that many factors acting at the individual and household level are responsible for low intention. All these factors are demand-side factors, but in conclusion, both demand-side and supply-side interventions have been highlighted, whereas the supply-side interventions and their implementations have not been studied or described, even using secondary sources.

Introduction:

The introduction of the article is very generic. The authors argued for a holistic study for deeper understanding but did not explain how this will be helpful in the context of the family planning situation for Uttar Pradesh. Also provide some literature (may not necessarily be of family planning but other BCC interventions) how the holistic studies became helpful in drawing the programmatic recommendations and the outcome of the studies showed evidence of better programmatic outcome or impact.

The authors listed the programmatic efforts at the state level to increase the uptake of FP services. But how much those efforts have been successful? Please provide some figures on programmatic inputs. Everything in this paper is about the users. However, at the same time, it is important to know whether programmatic inputs are enough or not. Otherwise, it will seem like all programmatic efforts are complete; but the uptake is not happening only due to issues in the demand-side.

Method:

How authors decided the content of in-depth or immersion interviews. Were those based on the previous literature or from stakeholders’ interviews? What was the theoretical framework, and how that fits within the existing FP program in the state? Otherwise, it will be difficult to draw programmatic recommendations out of this study.

Line 108: What does it mean by an immersion interview? How it is different from an in-depth interview? If methodologically not much difference between the two, suggesting using the term IDI because the use of that term is more popular.

Line 115: Please clarify what does it mean by ‘safe method’. Does it mean ‘safe-period method’ or ‘standard day method’, if so kindly use either of the two terms because those are more standardized terms?

Line 129: Remove the extra ‘period’ before (19).

Line 157: While providing the reference of a government document for high priority districts (reference #24) the authors wrote ‘Bhawan N’ as the author of the document. So far, my knowledge goes, it denotes “Nirman Bhawan”, which is a government administrative building in New Delhi. How can that be the author of a government document? Requesting authors to carefully check all references and avoid any such ambiguity and provide a proper citation for the said document. If any such document is available online, kindly provide the link for the readers in the reference as well.

Line 184: What does it mean by a 3-hour radius from state capital? I guess the authors mean a 3-hour driving distance. But that is a vague measurement. Suggesting providing an approximate distance or range in an absolute unit, kilometer.

Line 185: In place of ‘mixture’ suggesting ‘combination’.

Line 187-188. The authors selected the study block based on community literacy levels, the percentage of rural dwellers, and the percentage of community members coming from Scheduled Castes/Scheduled Tribes. The selection was based on whether the blocks had those indicators as high or low. Now the question comes what was considered as high and what as low. Please provide details in a table (maybe as an appendix) for those, otherwise it seems like the selection is highly purposive and not unbiased. For a qualitative study that is not a problem, but at least declares that it was a purposive selection if that was so. Also, which dataset provided the information for high and low percentages. I guess the authors took those from the census. If so, or anything else, please provide the reference.

Line 190-194: Kindly provide the reference of the data-set which have been used to calculate these indicators. Were these taken from NFHS or DLFPS?

Line 194-195: Please provide details of the methodology (maybe in the appendix) on how the authors determined what is ‘good’, ‘average’, or ‘poor’.

Results:

Line 259 Please update the reference.

Line 267: In NFHS-4 both SDM and LAM have been considered as modern methods, not the traditional method. What adjustments did the authors take to make NFHS-4 data comparable with other survey rounds? Since DLFPS data only came from 25 HPDs authors should recalculate the indicators only among 25 HPDs of UP for other survey rounds to make those comparable with DLFPS estimations.

Line 286: Please update the reference.

Line 304: Please update the citation in the text. Requesting authors to carefully check the pdf file, generated for submission, before approving the submission.

Line 320-322: I guess the authors changed the denominator in the second part of the sentence from the first part. Otherwise, it does not make a sense that “… 65% of women in the sample who wanted to limit, 79% intended to use any method, and 65% used any method…”. Please rephrase the whole sentence for both groups, who wanted to limit and to space.

Line 327: From the previous paragraph (line 318-326) it seems authors want to pitch for bridging the gap between awareness and intention to use. The gap between awareness and intention is wider than the gap between intention and use. Therefore, suggesting exploring the contextual factor for intention to use rather than the use of methods, and that could be an original contribution from this analysis.

Line 368: Please update the reference.

Line 419: Please update the reference.

Discussion:

Line 447 Are these factors responsible for low intention, or low use despite intentions?

Line 449 If fear of infertility is high why those are not getting adopted by those who want to limit?

Line 449: Low-risk perception of consequences of FP non-use or low-risk perception of having additional child, or low perception of avoiding the additional child using FP. Please clarify.

Line 450-453: In the Indian perspective there could be another pathway—those accept permanent method as their first FP method after reaching a certain number of children/sons. Requesting authors to consider this pathway as well.

Line 462: How the fear of infertility and side-effects remained unaddressed in the current intervention? Were not those parts of the intervention when the current intervention was planned or were there any challenges while implementing those?

Line 463: “Framing economic arguments for contraceptive use” could also raise some ethical issues as well. Raising a high number of children may put economic stress on the poor families but the economic argument may sound like it is ok for the ‘rich people’ to have as many children as they want because only they can afford it and poor families should have many children because they cannot afford!!! FP use is the right of the women and couples and the health program should make the methods accessible to those who need in a free, unbiased, and affordable way without any kind of coercion. Please reconsider the economic argument.

Line 479: Indian FP program does not include traditional methods so programmatically those methods do not get the endorsement. Why an intervention is required to discredit those methods? Kindly explain how discouraging the traditional methods are not going against the rights of the women or couples?

Line 485: Emphasizing the financial benefits of FP for leveraging men can also result in coercion against women’s will for adopting FP methods like sterilization. Therefore, how sensible it is to talk about financial benefits to influence the husband’s decision for FP use.

Line 488: Engaging the private sector is important for making the FP market more sustainable. I agree that the involvement of the private sector is important however, the authors' argument for private-sector engagement is not very convincing. Authors said because the drivers and barriers to contraceptive use are varied by person and method, FP offerings should adopt a private-sector approach. Not sure why a public sector approach will not be able to address those issues?

Also, how could a market-segmented approach simulate the uptake or stimulate the uptake? Please explain.

Line 497: What does it mean by ‘downward bias the coefficient’. Please explain.

Figures:

Please add a title to each figure.

Figure-1: Put X-axis on a time scale. Currently, it is categorial. Kindly justify why the results of the statewide surveys and the DLFPS are in the same graph. Did the estimates for NFHS and AHS have calculated only for those 25 HPDs? If not, requesting authors to do so, otherwise, remove the DLFPS estimates from the figure.

Figure-2: Requesting the authors to put the explanations of the cascade of 1, 2, and 3 as footnotes to the figure. I know the information is available in the text but a figure should be independent of the text—a stand-alone illustration.

Figure-3: This figure and the whole analysis is heavily tilted towards the demand side. I was wondering why the supply-side issues and quality of care were not considered within this FP journey mapping. If the state has an issue of unavailability and access to contraceptive methods, it can create a barrier to translate the intention to use to actual use. Also, the poor quality of care can result in distrust of the health system and that can aggravate the fear of side-effects and fear of infertility which further acts as a barrier. Therefore,

References:

Plos One uses the Vancouver style of referencing. Requesting the author to prepare the whole reference section following the style. Currently, there are several inconsistencies in the list of references. Some such inconsistencies are the following.

1. The journal names for reference #2, #3, and #18 were written differently. Even the same journal names have been differently formatted in different references e.g. #28 and #29.

2. The references of AHSs all stat with parenthesis and the author names or organization names such as OotRGaCC or IIfPSIaM is very hard to figure out. Requesting authors to provide enough and standard information, required for reference so that readers and future researchers can benefit from this article.

3. Please format the reference for NFHS properly, a suggested citation for NFHS is provided in every NFHS report.

6. PLOS authors have the option to publish the peer review history of their article (what does this mean?). If published, this will include your full peer review and any attached files.

Reviewer #1: **Yes: **Sanghamitra Singh

Reviewer #2: **Yes: **Arupendra Mozumdar

---

## [Author Response · Author response to Decision Letter 0]

27 Oct 2020

Thank you to the editor and the reviewers. We have responded to all comments received in our Response to Reviewers and made appropriate edits in the Revised Manuscript with Track Changes. We have also addressed the question of access to de-identified data and informed consent.

---

## [Decision Letter · Decision Letter 1]

10 Nov 2020

PONE-D-20-21461R1

Understanding drivers of family planning in rural northern India: an integrated mixed-methods approach

PLOS ONE

Dear Dr. Sgaier,

Thank you for submitting your manuscript to PLOS ONE. After careful consideration, we feel that it has merit but does not fully meet PLOS ONE’s publication criteria as it currently stands. Therefore, we invite you to submit a revised version of the manuscript that addresses the points raised during the review process.

The reviewers have suggested for minor amendments/corrections. The reviewers' comments are appended below. Kindly address those comments or suggestions.

We look forward to receiving your revised manuscript.

Kind regards,

Kannan Navaneetham, PhD

Academic Editor

PLOS ONE

Reviewers' comments:

Reviewer's Responses to Questions

**Comments to the Author**

1. If the authors have adequately addressed your comments raised in a previous round of review and you feel that this manuscript is now acceptable for publication, you may indicate that here to bypass the “Comments to the Author” section, enter your conflict of interest statement in the “Confidential to Editor” section, and submit your "Accept" recommendation.

Reviewer #1: All comments have been addressed

Reviewer #2: All comments have been addressed

2. Is the manuscript technically sound, and do the data support the conclusions?

Reviewer #1: Partly

Reviewer #2: Yes

3. Has the statistical analysis been performed appropriately and rigorously? 

Reviewer #1: I Don't Know

Reviewer #2: Yes

4. Have the authors made all data underlying the findings in their manuscript fully available?

Reviewer #1: Yes

Reviewer #2: No

5. Is the manuscript presented in an intelligible fashion and written in standard English?

Reviewer #1: Yes

Reviewer #2: Yes

6. Review Comments to the Author

Reviewer #1: The article examines the important factors influencing demand generation in family planning. While the findings are relevant, some questions remain unanswered and some over-simplistic assumptions seem to have been made while analyzing the data. For example:

Lines 382-383: Talks about contraceptive use by religion. Have the authors tried to look at contraceptive uptake across religions AND wealth quintiles as well as literacy rates. This is a very important aspect that should be analyzed here.

Line 456 onwards: Were questions around risk perception on contraceptive side effects accompanied by questions around access to quality family planning counselling services? Again this is very important to understand. In fact the discussion section should also talk about provider bias which dissuades users from accessing contraceptive services, specially young people.

Overall: There should be a perspective on Uttar Pradesh's large young population, particularly the large adolescent population. In fact the state has the largest adolescent population in India. Their contraceptive needs are distinct, which also need to be catered to and the manuscript should include a mention of that.

Reviewer #2: Thank you for giving me the opportunity to review the revised version of the article titled “Understanding drivers of family planning in rural northern India: an integrated mixed-methods approach” (PONE-D-20-21461R1). I also thank the authors for considering the comments in revising the manuscript. Please find some minor comments on this version of the manuscript. Hope those will be helpful.

Specific comments: -

Introduction:

Line 114 The governments in India have taken a target-free approach for implementing the FP program. Therefore, suggesting replacing the word ‘target’ from this sentence with another suitable word.

Method:

Line 194 Requesting to add a short note on ASHA areas, maybe as a footnote, especially for the readers from outside India.

Line 210 Suggesting replacing ‘simple chi-square test’ with ‘chi-square test of independence’.

Results:

Line 345 Kindly include male sterilization among the list of modern contraceptives.

Line 386 Suggesting rephrasing “Awareness of other methods was a negative predictor of intention to use female sterilization, and condoms.” I am not questioning the association, but awareness of other methods is probably not predicting the non-use of sterilization and condoms, it is something else. It is more possible that users of those two methods do not know any other methods. It raises the question of the quality of care received by those women; they were not told about other methods by the providers at the time of method adoption at least in the case of sterilization.

Line 389 I don’t understand what does it mean by “perceived access to be hard for sterilization”. Suggesting the rephrase in a simpler language.

Line 396 Suggesting using “demographic characteristics” instead of “demographic mix”.

Line 418 If women don’t have any power over the decision of condom use, it is a problem of women’s rights. What is the logic behind reporting the decision making of condom use as ‘hierarchical’? I am sorry if I misunderstood, but the use of the word ‘hierarchical’ in some way sounds like a justification for lack of decision-making power, as if a woman’s status is lower to her husband or other family members, and it’s like everyone (at least the people of the scientific community) all accept it that way. Do authors themselves perceive husbands are in a higher social status than women? If not, I suggest just report who is making the decision for condom use. If it is the husband or in-laws or any other family members, just report that without any justification, such that, it does not appear as if it is an obvious thing to happen. It is not acceptable in any condition that women are not in a position to make her own decision for using a contraceptive.

Line 477 I would like to say this sentence about the usefulness of financial risk perception as a target for intervention should go to the discussion section of the paper leaving the result section only for findings and not to draw any programmatic recommendation in the result section.

Discussion:

Line 551 I agree that financial issue plays a big role in decision making of the couples especially the husbands on their fertility preference. But as I pointed out in my original review bringing the risk of financial issues in interventions could also raise some ethical issues. Putting the financial argument in program intervention for contraceptive use will disproportionately discourage the families of low socioeconomic strata to achieve their fertility goal. Therefore, I suggest that the authors should draw programmatic recommendations by addressing the consequences of financial risk on family health instead of including the issues of financial risk directly.

Line 564 It depends on which model we are talking about. If we are dealing with logistic models shifting the coefficient towards 0 is not a bad thing. Suggesting changing “…the coefficients towards 0” into “…the coefficients towards non-significance”.

Figures:

Please add a title to each figure.

Could not find figure 1.

References:

1. Please provide more details for Ref#5. If it is a website, please provide the URL and the date when it was accessed for the last time.

2. Please provide more details for Ref#14. If it is a report available on a website, please provide the URL and date when it was accessed for the last time.

3. Please correct the author's name of the Ref#16.

4. Please correct the NFHS-4 reference (Ref#24). In the author list, Macro international should be replaced by ICF. Need to add the same in Ref#16 as well.

7. PLOS authors have the option to publish the peer review history of their article (what does this mean?). If published, this will include your full peer review and any attached files.

Reviewer #1: **Yes: **Sanghamitra Singh, Population Foundation of India

Reviewer #2: **Yes: **Arupendra Mozumdar

---

## [Author Response · Author response to Decision Letter 1]

25 Nov 2020

Thank you for your comments - we truly believe they have strengthened our paper. We have addressed all comments in the revised version of the manuscript and addressed the reviews comments in the new response to reviewers document.

---

## [Editor Report · Decision Letter 2]

30 Nov 2020

Understanding drivers of family planning in rural northern India: an integrated mixed-methods approach

PONE-D-20-21461R2

Dear Dr. Sgaier,

We’re pleased to inform you that your manuscript has been judged scientifically suitable for publication and will be formally accepted for publication once it meets all outstanding technical requirements.

Kind regards,

Kannan Navaneetham, PhD

Academic Editor

PLOS ONE
---

## [Editor Report · Acceptance letter]

21 Dec 2020

PONE-D-20-21461R2 

Understanding drivers of family planning in rural northern India: an integrated mixed-methods approach 

Dear Dr. Sgaier:

I'm pleased to inform you that your manuscript has been deemed suitable for publication in PLOS ONE. Congratulations! Your manuscript is now with our production department. 

Kind regards, 

on behalf of

Professor Kannan Navaneetham 

Academic Editor

PLOS ONE